# ADVERSARIALLY GUIDED ACTOR-CRITIC

**Yannis Flet-Berliac**[*]
Inria, Scool team
Univ. Lille, CRIStAL, CNRS
`yannis.flet-berliac@inria.fr`

**Johan Ferret**[*]
Google Research, Brain team
Inria, Scool team
Univ. Lille, CRIStAL, CNRS

**Olivier Pietquin**
Google Research, Brain team

**Philippe Preux**
Inria, Scool team
Univ. Lille, CRIStAL, CNRS

**Matthieu Geist**
Google Research, Brain team

## ABSTRACT

Despite definite success in deep reinforcement learning problems, actor-critic
algorithms are still confronted with sample inefficiency in complex environments,
particularly in tasks where efficient exploration is a bottleneck. These methods
consider a policy (the actor) and a value function (the critic) whose respective losses
are built using different motivations and approaches. This paper introduces a third
protagonist: the adversary. While the adversary mimics the actor by minimizing the
KL-divergence between their respective action distributions, the actor, in addition to
learning to solve the task, tries to differentiate itself from the adversary predictions.
This novel objective stimulates the actor to follow strategies that could not have
been correctly predicted from previous trajectories, making its behavior innovative
in tasks where the reward is extremely rare. Our experimental analysis shows that
the resulting Adversarially Guided Actor-Critic (`AGAC`) algorithm leads to more
exhaustive exploration. Notably, `AGAC` outperforms current state-of-the-art methods
on a set of various hard-exploration and procedurally-generated tasks.

## 1 INTRODUCTION

Research in deep reinforcement learning (RL) has proven to be successful across a wide range
of problems (Silver et al., 2014; Schulman et al., 2016; Lillicrap et al., 2016; Mnih et al., 2016).
Nevertheless, generalization and exploration in RL still represent key challenges that leave most
current methods ineffective. First, a battery of recent studies (Farebrother et al., 2018; Zhang et al.,
2018a; Song et al., 2020; Cobbe et al., 2020) indicates that current RL methods fail to generalize
correctly even when agents have been trained in a diverse set of environments. Second, exploration has
been extensively studied in RL; however, most hard-exploration problems use the same environment
for training and evaluation. Hence, since a well-designed exploration strategy should maximize the
information received from a trajectory about an environment, the exploration capabilities may not be
appropriately assessed if that information is memorized. In this line of research, we choose to study
the exploration capabilities of our method and its ability to generalize to new scenarios. Our evaluation
domains will, therefore, be tasks with sparse reward in procedurally-generated environments.

In this work, we propose Adversarially Guided Actor-Critic (`AGAC`), which reconsiders the actor-critic
framework by introducing a third protagonist: the adversary. Its role is to predict the actor's actions
correctly. Meanwhile, the actor must not only find the optimal actions to maximize the sum of
expected returns, but also counteract the predictions of the adversary. This formulation is lightly
inspired by adversarial methods, specifically generative adversarial networks (GANs) (Goodfellow
et al., 2014). Such a link between GANs and actor-critic methods has been formalized by Pfau &
Vinyals (2016); however, in the context of a third protagonist, we draw a different analogy. The
adversary can be interpreted as playing the role of a discriminator that must predict the actions of the
actor, and the actor can be considered as playing the role of a generator that behaves to deceive the
predictions of the adversary. This approach has the advantage, as with GANs, that the optimization
procedure generates a diversity of meaningful data, corresponding to sequences of actions in `AGAC`.

---

[*]Equal contribution.

This paper analyses and explores how `AGAC` explicitly drives diversity in the behaviors of the agent while remaining reward-focused, and to which extent this approach allows to adapt to the evolving state space of procedurally-generated environments where the map is constructed differently with each new episode. Moreover, because stability is a legitimate concern since specific instances of adversarial networks were shown to be prone to hyperparameter sensitivity issues (Arjovsky & Bottou, 2017), we also examine this aspect in our experiments.

The contributions of this work are as follow: (i) we propose a novel actor-critic formulation inspired from adversarial learning (`AGAC`), (ii) we analyse empirically `AGAC` on key reinforcement learning aspects such as diversity, exploration and stability, (iii) we demonstrate significant gains in performance on several sparse-reward hard-exploration tasks including procedurally-generated tasks.

## 2 RELATED WORK

Actor-critic methods (Barto et al., 1983; Sutton, 1984) have been extended to the deep learning setting by Mnih et al. (2016), who combined deep neural networks and multiple distributed actors with an actor-critic setting, with strong results on Atari. Since then, many additions have been proposed, be it architectural improvements (Vinyals et al., 2019), better advantage or value estimation (Schulman et al., 2016; Flet-Berliac et al., 2021), or the incorporation of off-policy elements (Wang et al., 2017; Oh et al., 2018; Flet-Berliac & Preux, 2020). Regularization was shown to improve actor-critic methods, either by enforcing trust regions (Schulman et al., 2015; 2017; Wu et al., 2017), or by correcting for off-policiness (Munos et al., 2016; Gruslys et al., 2018); and recent works analyzed its impact from a theoretical standpoint (Geist et al., 2019; Ahmed et al., 2019; Vieillard et al., 2020a;b). Related to our work, Han & Sung (2020) use the entropy of the mixture between the policy induced from a replay buffer and the current policy as a regularizer. To the best of our knowledge, none of these methods explored the use of an adversarial objective to drive exploration.

While introduced in supervised learning, adversarial learning (Goodfellow et al., 2015; Miyato et al., 2016; Kurakin et al., 2017) was leveraged in several RL works. Ho & Ermon (2016) propose an imitation learning method that uses a discriminator whose task is to distinguish between expert trajectories and those of the agent while the agent tries to match expert behavior to fool the discriminator. Bahdanau et al. (2019) use a discriminator to distinguish goal states from non-goal states based on a textual instruction, and use the resulting model as a reward function. Florensa et al. (2018) use a GAN to produce sub-goals at the right level of difficulty for the current agent, inducing a form of curriculum. Additionally, Pfau & Vinyals (2016) provide a parallel between GANs and the actor-critic framework.

While exploration is driven in part by the core RL algorithms (Fortunato et al., 2018; Han & Sung, 2020; Ferret et al., 2021), it is often necessary to resort to exploration-specific techniques. For instance, intrinsic motivation encourages exploratory behavior from the agent. Some works use state-visitation counts or pseudo-counts to promote exhaustive exploration (Bellemare et al., 2016a), while others use curiosity rewards, expressed in the magnitude of prediction error from the agent, to push it towards unfamiliar areas of the state space (Burda et al., 2018). Ecoffet et al. (2019) propose a technique akin to tree traversal to explore while learning to come back to promising areas. Eysenbach et al. (2018) show that encouraging diversity helps with exploration, even in the absence of reward.

Last but not least, generalization is a key challenge in RL. Zhang et al. (2018b) showed that, even when the environment is not deterministic, agents can overfit to their training distribution and that it is difficult to distinguish agents likely to generalize to new environments from those that will not. In the same vein, recent work has advocated using procedurally-generated environments, in which a new instance of the environment is sampled when a new episode starts, to assess generalization capabilities better (Justesen et al., 2018; Cobbe et al., 2020). Finally, methods based on network randomization (Igl et al., 2019), noise injection (Lee et al., 2020), and credit assignment (Ferret et al., 2020) have been proposed to reduce the generalization gap for RL agents.

## 3 BACKGROUND AND NOTATIONS

We place ourselves in the Markov Decision Processes (Puterman, 1994) framework. A Markov Decision Process (MDP) is a tuple $M = \{S, A, P, R, \gamma\}$, where $S$ is the state space, $A$ is the action

space, $\mathcal{P}$ is the transition kernel, $\mathcal{R}$ is the bounded reward function and $\gamma \in [0, 1)$ is the discount factor. Let $\pi$ denote a stochastic policy mapping states to distributions over actions. We place ourselves in the infinite-horizon setting, *i.e.*, we seek a policy that optimizes $J(\pi) = \mathbb{E}_\pi[\sum_{t=0}^{\infty} \gamma^t r(s_t, a_t)]$. The value of a state is the quantity $V^\pi(s) = \mathbb{E}_\pi[\sum_{t=0}^{\infty} \gamma^t r(s_t, a_t) | s_0 = s]$ and the value of a state-action pair $Q^\pi(s, a)$ of performing action $a$ in state $s$ and then following policy $\pi$ is defined as: $Q^\pi(s, a) = \mathbb{E}_\pi[\sum_{t=0}^{\infty} \gamma^t r(s_t, a_t) | s_0 = s, a_0 = a]$. The advantage function, which quantifies how an action $a$ is better than the average action in state $s$, is $A^\pi(s, a) = Q^\pi(s, a) - V^\pi(s)$. Finally, the entropy $\mathcal{H}^\pi$ of a policy is calculated as: $\mathcal{H}^\pi(s) = \mathbb{E}_{\pi(\cdot|s)}[-\log \pi(\cdot|s)]$.

**Actor-Critic and Deep Policy Gradients.** An actor-critic algorithm is composed of two main components: a policy and a value predictor. In deep RL, both the policy and the value function are obtained via parametric estimators; we denote $\theta$ and $\phi$ their respective parameters. The policy is updated via policy gradient, while the value is usually updated via temporal difference or Monte Carlo rollouts. In practice, for a sequence of transitions $\{s_t, a_t, r_t, s_{t+1}\}_{t \in [0,N]}$, we use the following policy gradient loss (including the commonly used entropic penalty):

$$\mathcal{L}_{PG} = -\frac{1}{N} \sum_{t'=t}^{t+N} (A_{t'} \log \pi(a_{t'}|s_{t'}, \theta) + \alpha \mathcal{H}^\pi(s_{t'}, \theta)),$$

where $\alpha$ is the entropy coefficient and $A_t$ is the generalized advantage estimator (Schulman et al., 2016) defined as: $A_t = \sum_{t'=t}^{t+N} (\gamma\lambda)^{t'-t}(r_{t'} + \gamma V_{\phi_{\text{old}}}(s_{t'+1}) - V_{\phi_{\text{old}}}(s_{t'}))$, with $\lambda$ a fixed hyperparameter and $V_{\phi_{\text{old}}}$ the value function estimator at the previous optimization iteration. To estimate the value function, we solve the non-linear regression problem $\text{minimize}_\phi \sum_{t'=t}^{t+N}(V_\phi(s_{t'}) - \hat{V}_{t'})^2$ where $\hat{V}_t = A_t + V_{\phi_{\text{old}}}(s_{t'})$.

## 4 ADVERSARIALLY GUIDED ACTOR-CRITIC

To foster diversified behavior in its trajectories, AGAC introduces a third protagonist to the actor-critic framework: the adversary. The role of the adversary is to accurately predict the actor's actions, by *minimizing* the discrepancy between its action distribution $\pi_{\text{adv}}$ and the distribution induced by the policy $\pi$. Meanwhile, in addition to finding the optimal actions to *maximize* the sum of expected returns, the actor must also counteract the adversary's predictions by *maximizing* the discrepancy between $\pi$ and $\pi_{\text{adv}}$ (see Appendix B for an illustration). This discrepancy, used as a form of exploration bonus, is defined as the difference of action log-probabilities (see Eq. (1)), whose expectation is the Kullback–Leibler divergence:

$$D_{\text{KL}}(\pi(\cdot|s)\|\pi_{\text{adv}}(\cdot|s)) = \mathbb{E}_{\pi(\cdot|s)}[\log \pi(\cdot|s) - \log \pi_{\text{adv}}(\cdot|s)].$$

Formally, for each state-action pair $(s_t, a_t)$ in a trajectory, an action-dependent bonus $\log \pi(a_t|s_t) - \log \pi_{\text{adv}}(a_t|s_t)$ is added to the advantage. In addition, the value target of the critic is modified to include the action-independent equivalent, which is the KL-divergence $D_{\text{KL}}(\pi(\cdot|s_t)\|\pi_{\text{adv}}(\cdot|s_t))$. We discuss the role of these mirrored terms below, and the implications of AGAC's modified objective from a more theoretical standpoint in the next section. In addition to the parameters $\theta$ (resp. $\theta_{\text{old}}$ the parameter of the policy at the previous iteration) and $\phi$ defined above (resp. $\phi_{\text{old}}$ that of the critic), we denote $\psi$ (resp. $\psi_{\text{old}}$) that of the adversary.

AGAC minimizes the following loss:

$$\mathcal{L}_{\text{AGAC}} = \mathcal{L}_{\text{PG}} + \beta_V \mathcal{L}_V + \beta_{\text{adv}} \mathcal{L}_{\text{adv}}.$$

In the new objective $\mathcal{L}_{PG} = -\frac{1}{N} \sum_{t=0}^{N} (A_t^{\text{AGAC}} \log \pi(a_t|s_t, \theta) + \alpha \mathcal{H}^\pi(s_t, \theta))$, AGAC modifies $A_t$ as:

$$A_t^{\text{AGAC}} = A_t + c\left(\log \pi(a_t|s_t, \theta_{\text{old}}) - \log \pi_{\text{adv}}(a_t|s_t, \psi_{\text{old}})\right), \tag{1}$$

with $c$ a varying hyperparameter that controls the dependence on the action log-probability difference. To encourage exploration without preventing asymptotic stability, $c$ is linearly annealed during the course of training. $\mathcal{L}_V$ is the objective function of the critic defined as:

$$\mathcal{L}_V = \frac{1}{N} \sum_{t=0}^{N} \left(V_\phi(s_t) - \left(\hat{V}_t + c\, D_{\text{KL}}(\pi(\cdot|s_t, \theta_{\text{old}})\|\pi_{\text{adv}}(\cdot|s_t, \psi_{\text{old}}))\right)\right)^2. \tag{2}$$

Finally, $\mathcal{L}_{\text{adv}}$ is the objective function of the adversary:

$$\mathcal{L}_{\text{adv}} = \frac{1}{N} \sum_{t=0}^{N} D_{\text{KL}}(\pi(\cdot|s_t, \theta_{\text{old}})\|\pi_{\text{adv}}(\cdot|s_t, \psi)). \tag{3}$$

Eqs. (1), (2) and (3) are the three equations that our method modifies (we color in blue the specific parts) in the traditional actor-critic framework. The terms $\beta_V$ and $\beta_{\text{adv}}$ are fixed hyperparameters.

Under the proposed actor-critic formulation, the probability of sampling an action is increased if the modified advantage is positive, *i.e.* (i) the corresponding return is larger than the predicted value and/or (ii) the action log-probability difference is large. More precisely, our method favors transitions whose actions were less accurately predicted than the average action, *i.e.* $\log \pi(a|s) - \log \pi_{\text{adv}}(a|s) \geq D_{\text{KL}}(\pi(\cdot|s)\|\pi_{\text{adv}}(\cdot|s))$. This is particularly visible for $\lambda \to 1$, in which case the generalized advantage is $A_t = G_t - V_{\phi_{\text{old}}}(s_t)$, resulting in the appearance of both aforementioned mirrored terms in the modified advantage:

$$A_t^{\text{AGAC}} = G_t - \hat{V}_t^{\phi_{\text{old}}} + c \left( \log \pi(a_t|s_t) - \log \pi_{\text{adv}}(a_t|s_t) - \hat{D}_{\text{KL}}^{\phi_{\text{old}}}(\pi(\cdot|s_t)\|\pi_{\text{adv}}(\cdot|s_t)) \right),$$

with $G_t$ the observed return, $\hat{V}_t^{\phi_{\text{old}}}$ the estimated return and $\hat{D}_{\text{KL}}^{\phi_{\text{old}}}(\pi(\cdot|s_t)\|\pi_{\text{adv}}(\cdot|s_t))$ the estimated KL-divergence (estimated components of $V_{\phi_{\text{old}}}(s_t)$ from Eq. 2).

To avoid instability, in practice the adversary is a separate estimator, updated with a smaller learning rate than the actor. This way, it represents a delayed and more steady version of the actor's policy, which prevents the agent from having to constantly adapt or focus solely on fooling the adversary.

## 4.1 BUILDING MOTIVATION

In the following, we provide an interpretation of AGAC by studying the dynamics of attraction and repulsion between the actor and the adversary. To simplify, we study the equivalent of AGAC in a policy iteration (PI) scheme. PI being the dynamic programming scheme underlying the standard actor-critic, we have reasons to think that some of our findings translate to the original AGAC algorithm. In PI, the quantity of interest is the action-value, which AGAC would modify as:

$$Q_{\pi_k}^{\text{AGAC}} = Q_{\pi_k} + c \left( \log \pi_k - \log \pi_{\text{adv}} \right),$$

with $\pi_k$ the policy at iteration $k$. Incorporating the entropic penalty, the new policy $\pi_{k+1}$ verifies:

$$\pi_{k+1} = \arg\max_{\pi} \mathcal{J}_{\text{PI}}(\pi) = \arg\max_{\pi} \mathbb{E}_s \mathbb{E}_{a \sim \pi(\cdot|s)}[Q_{\pi_k}^{\text{AGAC}}(s, a) - \alpha \log \pi(a|s)].$$

We can rewrite this objective:

$$\begin{aligned}
\mathcal{J}_{\text{PI}}(\pi) &= \mathbb{E}_s \mathbb{E}_{a \sim \pi(\cdot|s)}[Q_{\pi_k}^{\text{AGAC}}(s, a) - \alpha \log \pi(a|s)] \\
&= \mathbb{E}_s \mathbb{E}_{a \sim \pi(\cdot|s)}[Q_{\pi_k}(s, a) + c \left( \log \pi_k(a|s) - \log \pi_{\text{adv}}(a|s) \right) - \alpha \log \pi(a|s)] \\
&= \mathbb{E}_s \mathbb{E}_{a \sim \pi(\cdot|s)}[Q_{\pi_k}(s, a) + c \left( \log \pi_k(a|s) - \log \pi(a|s) + \log \pi(a|s) - \log \pi_{\text{adv}}(a|s) \right) - \alpha \log \pi(a|s)] \\
&= \mathbb{E}_s \left[ \mathbb{E}_{a \sim \pi(\cdot|s)}[Q_{\pi_k}(s, a)] \underbrace{- c\,D_{\text{KL}}(\pi(\cdot|s)\|\pi_k(\cdot|s))}_{\pi_k \text{ is attractive}} \underbrace{+ c\,D_{\text{KL}}(\pi(\cdot|s)\|\pi_{\text{adv}}(\cdot|s))}_{\pi_{\text{adv}} \text{ is repulsive}} \underbrace{+ \alpha\,\mathcal{H}(\pi(\cdot|s))}_{\text{enforces stochastic policies}} \right].
\end{aligned}$$

Thus, in the PI scheme, AGAC finds a policy that maximizes $Q$-values, while at the same time remaining close to the current policy and far from a mixture of the previous policies (*i.e.*, $\pi_{k-1}, \pi_{k-2}, \pi_{k-3}, \ldots$). Note that we experimentally observe (see Section 5.3) that our method performs better with a smaller learning rate for the adversarial network than that of the other networks, which could imply that a stable repulsive term is beneficial.

This optimization problem is strongly concave in $\pi$ (thanks to the entropy term), and is state-wise a Legendre-Fenchel transform. Its solution is given by (see Appendix E for the full derivation):

$$\pi_{k+1} \propto \left( \frac{\pi_k}{\pi_{\text{adv}}} \right)^{\frac{c}{\alpha}} \exp \frac{Q_{\pi_k}}{\alpha}.$$

This result gives us some insight into the behavior of the objective function. Notably, in our example, if $\pi_{\text{adv}}$ is fixed and $c = \alpha$, we recover a KL-regularized PI scheme (Geist et al., 2019) with the modified reward $r - c \log \pi_{\text{adv}}$.

## 4.2 IMPLEMENTATION

In all of the experiments, we use PPO (Schulman et al., 2017) as the base algorithm and build on it to incorporate our method. Hence,

$$\mathcal{L}_{PG} = -\frac{1}{N}\sum_{t'=t}^{t+N} \min\left(\frac{\pi(a_{t'}|s_{t'},\theta)}{\pi(a_{t'}|s_{t'},\theta_{\text{old}})}A_{t'}^{\texttt{AGAC}}, \text{clip}\left(\frac{\pi(a_{t'}|s_{t'},\theta)}{\pi(a_{t'}|s_{t'},\theta_{\text{old}})}, 1-\epsilon, 1+\epsilon\right)A_{t'}^{\texttt{AGAC}}\right),$$

with $A_{t'}^{\texttt{AGAC}}$ given in Eq. (1), $N$ the temporal length considered for one update of parameters and $\epsilon$ the clipping parameter. Similar to RIDE (Raileanu & Rocktäschel, 2019), we also discount PPO by episodic state visitation counts, except for VizDoom (*cf.* Section 5.1). The actor, critic and adversary use the convolutional architecture of the Nature paper of DQN (Mnih et al., 2015) with different hidden sizes (see Appendix D for architecture details). The three neural networks are optimized using Adam (Kingma & Ba, 2015). Our method does not use RNNs in its architecture; instead, in all our experiments, we use frame stacking. Indeed, Hausknecht & Stone (2015) interestingly demonstrate that although recurrence is a reliable method for processing state observation, it does not confer any systematic advantage over stacking observations in the input layer of a CNN. Note that the parameters are not shared between the policy, the critic and the adversary and that we did not observe any noticeable difference in computational complexity when using AGAC compared to PPO. We direct the reader to Appendix C for a list of hyperparameters. In particular, the $c$ coefficient of the adversarial bonus is linearly annealed.

At each training step, we perform a stochastic optimization step to minimize $\mathcal{L}_{\texttt{AGAC}}$ using stop-gradient:

$$\theta \leftarrow \text{Adam}\left(\theta, \nabla_\theta \mathcal{L}_{PG}, \eta_1\right), \qquad \phi \leftarrow \text{Adam}\left(\phi, \nabla_\phi \mathcal{L}_V, \eta_1\right), \qquad \psi \leftarrow \text{Adam}\left(\psi, \nabla_\psi \mathcal{L}_{\text{adv}}, \eta_2\right).$$

## 5 EXPERIMENTS

In this section, we describe our experimental study in which we investigate: (i) whether the adversarial bonus alone (*e.g.* without episodic state visitation count) is sufficient to outperform other methods in VizDoom, a sparse-reward task with high-dimensional observations, (ii) whether AGAC succeeds in partially-observable and procedurally-generated environments with high sparsity in the rewards, compared to other methods, (iii) how well AGAC is capable of exploring in environments without extrinsic reward, (iv) the training stability of our method. In all of the experiments, lines are average performances and shaded areas represent one standard deviation. The code for our method is released at github.com/yfletberliac/adversarially-guided-actor-critic.

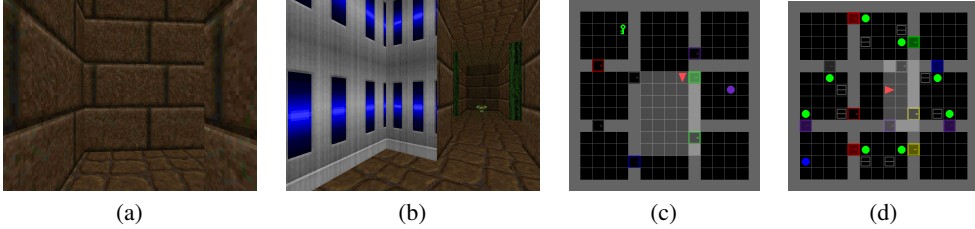

Figure 1: (a,b) Frames from the 3-D navigation task VizdoomMyWayHome. (c) MiniGrid-KeyCorridorS6R3. (d) MiniGrid-ObstructedMazeFull.

**Environments.** To carefully evaluate the performance of our method, its ability to develop robust exploration strategies and its generalization to unseen states, we choose tasks that have been used in prior work, which are tasks with high-dimensional observations, sparse reward and procedurally-generated environments. In **VizDoom** (Kempka et al., 2016), the agent must learn to move along corridors and through rooms without any reward feedback from the 3-D environment. The **MiniGrid** environments (Chevalier-Boisvert et al., 2018) are a set of challenging partially-observable and sparse-reward gridworlds. In this type of procedurally-generated environments, memorization is impossible due to the huge size of the state space, so the agent must learn to generalize across the different layouts of the environment. Each gridworld has different characteristics: in the MultiRoom

tasks, the agent is placed in the first room and should reach a goal placed in the most distant room. In the KeyCorridor tasks, the agent must navigate to pick up an object placed in a room locked by a door whose key is in another room. Finally, in the ObstructedMaze tasks, the agent must pick up a box that is placed in a corner of a 3x3 maze in which the doors are also locked, the keys are hidden in boxes and balls obstruct the doors. All considered environments (see Fig. 1 for some examples) are available as part of OpenAI Gym (Brockman et al., 2016).

**Baselines.** For a fair assessment of our method, we compare to some of the most prominent methods specialized in hard-exploration tasks: **RIDE** (Raileanu & Rocktäschel, 2019), based on an intrinsic reward associated with the magnitude of change between two consecutive state representations and state visitation, **Count** as Count-Based Exploration (Bellemare et al., 2016b), which we couple with IMPALA (Espeholt et al., 2018), **RND** (Burda et al., 2018) in which an exploration bonus is positively correlated to the error of predicting features from the observations and **ICM** (Pathak et al., 2017), where a module only predicts the changes in the environment that are produced by the actions of the agent. Finally, we compare to most the recent and best performing method at the time of writing in procedurally-generated environments: **AMIGo** (Campero et al., 2021) in which a goal-generating teacher provides count-based intrinsic goals.

## 5.1 ADVERSARIALLY-BASED EXPLORATION (NO EPISODIC COUNT)

Table 1: Average return in VizDoom at different timesteps.

| Nb. of Timesteps | 2M | 4M | 6M | 8M | 10M |
|---|---|---|---|---|---|
| **AGAC** | $\mathbf{0.74} \pm 0.05$ | $\mathbf{0.96} \pm 0.001$ | $\mathbf{0.96} \pm 0.001$ | $\mathbf{0.97} \pm 0.001$ | $\mathbf{0.97} \pm 0.001$ |
| RIDE | 0. | 0. | $0.95 \pm 0.001$ | $\mathbf{0.97} \pm 0.001$ | $\mathbf{0.97} \pm 0.001$ |
| ICM | 0. | 0. | $0.95 \pm 0.001$ | $\mathbf{0.97} \pm 0.001$ | $\mathbf{0.97} \pm 0.001$ |
| AMIGo | 0. | 0. | 0. | 0. | 0. |
| RND | 0. | 0. | 0. | 0. | 0. |
| Count | 0. | 0. | 0. | 0. | 0. |

In this section, we assess the benefits of using an adversarially-based exploration bonus and examine how `AGAC` performs without the help of count-based exploration. In order to provide a comparison to state-of-the-art methods, we choose VizDoom, a hard-exploration problem used in prior work. In this game, the map consists of 9 rooms connected by corridors where 270 steps separate the initial position of the agent and the goal under an optimal policy. Episodes are terminated either when the agent finds the goal or if the episode exceeds 2100 timesteps. Importantly, while other algorithms (Raileanu & Rocktäschel, 2019; Campero et al., 2021) benefit from count-based exploration, this study has been conducted with our method not benefiting from episodic count whatsoever. Results in Table 1 indicate that `AGAC` clearly outperforms other methods in sample-efficiency. Only the methods ICM and RIDE succeed in matching the score of `AGAC`, and with about twice as much transitions ($\sim$ 3M vs. 6M). Interestingly, AMIGo performs similarly to Count and RND. We find this result surprising because AMIGo has proven to perform well in the MiniGrid environments. Nevertheless, it appears that concurrent works to ours experienced similar issues with the accompanying implementation[1]. The results of `AGAC` support the capabilities of the adversarial bonus and show that it can, on its own, achieve significant gains in performance. However, the VizDoom task is not procedurally-generated; hence we have not evaluated the generalization to new states yet. In the following section, we use MiniGrid to investigate this.

## 5.2 HARD-EXPLORATION TASKS WITH PARTIALLY-OBSERVABLE ENVIRONMENTS

We now evaluate our method on multiple hard-exploration procedurally-generated tasks from Mini-Grid. Details about MiniGrid can be found in Appendix C.1. Fig. 2 indicates that `AGAC` significantly outperforms other methods on these tasks in sample-efficiency and performance. `AGAC` also outperforms the current state-of-the-art method, AMIGo, despite the fact that it uses the fully-observable version of MiniGrid. Note that we find the same poor performance results when training AMIGo in MiniGrid, similar to Vizdoom results. For completeness, we also report in Table 2 of Appendix A.1 the performance results with the scores reported in the original papers Raileanu & Rocktäschel

---

[1]AMIGo implementation GitHub Issue.

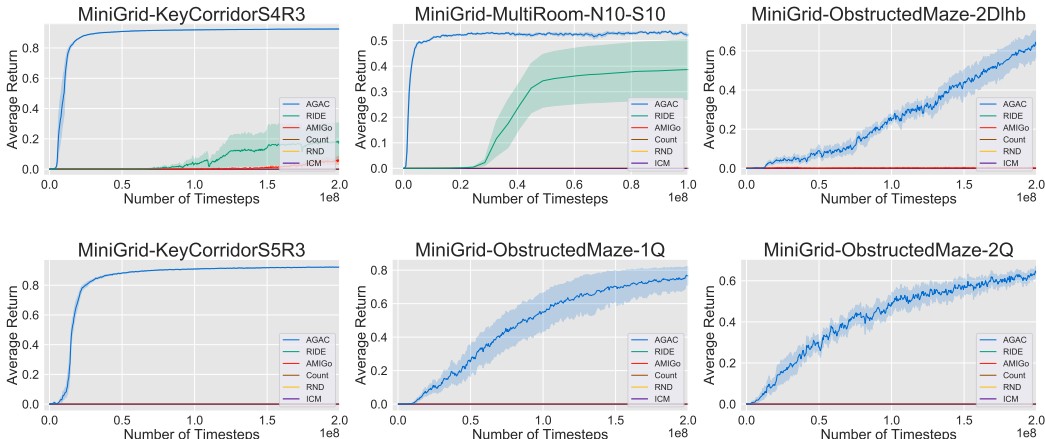

Figure 2: Performance evaluation of `AGAC`.

(2019) and Campero et al. (2021). We draw similar conclusions: `AGAC` clearly outperforms the state-of-the-art RIDE, AMIGo, Count, RND and ICM.

In all the considered tasks, the agent must learn to generalize across a very large state space because the layouts are generated procedurally. We consider three main arguments to explain why our method is successful: (i) our method makes use of partial observations: in this context, the adversary has a harder time predicting the actor's actions; nevertheless, the mistakes of the former benefit the latter in the form of an exploration bonus, which pushes the agent to explore further in order to deceive the adversary, (ii) the exploration bonus (*i.e.* intrinsic reward) does not dissipate compared to most other methods, as observed in Fig. 9 in Appendix A.4, (iii) our method does not make assumptions about the environment dynamics (*e.g.*, changes in the environment produced by an action as in Raileanu & Rocktäschel (2019)) since this can hinder learning when the space of state changes induced by an action is too large (such as the action of moving a block in ObstructedMaze).

In Appendix A.3, we also include experiments in two environments with extremely sparse reward signals: KeyCorridorS8R3 and ObstructedMazeFull. Despite the challenge, `AGAC` still manages to find rewards and can perform well by taking advantage of the diversified behaviour induced by our method. To the best of our knowledge, no other method ever succeeded to perform well ($> 0$ average return) in those tasks. We think that given more computing time, `AGAC`'s score could go higher.

## 5.3 TRAINING STABILITY

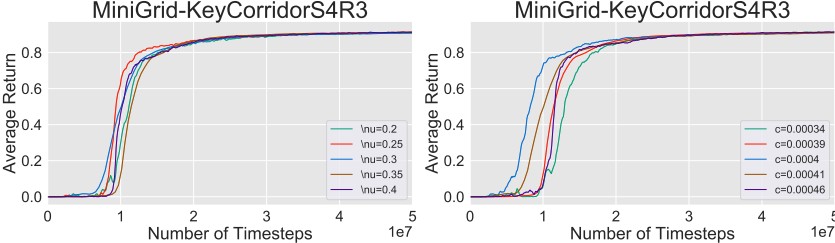

Figure 3: Sensitivity analysis of `AGAC` in KeyCorridorS4R3.

Here we want to analyse the stability of the method when changing hyperparameters. The most important parameters in `AGAC` are $c$, the coefficient for the adversarial bonus, and the learning rates ratio $\nu = \frac{\eta_2}{\eta_1}$. We choose KeyCorridorS4R3 as the evaluation task because among all the tasks considered, its difficulty is at a medium level. Fig. 3 shows the learning curves. For readability, we plot the average return only; the standard deviation is sensibly the same for all curves. We observe that deviating from the hyperparameter values found using grid search results in a slower training. Moreover, although reasonable, $c$ appears to have more sensitivity than $\nu$.

**RND**  **Count**  **Random**  **RIDE**  **AGAC**

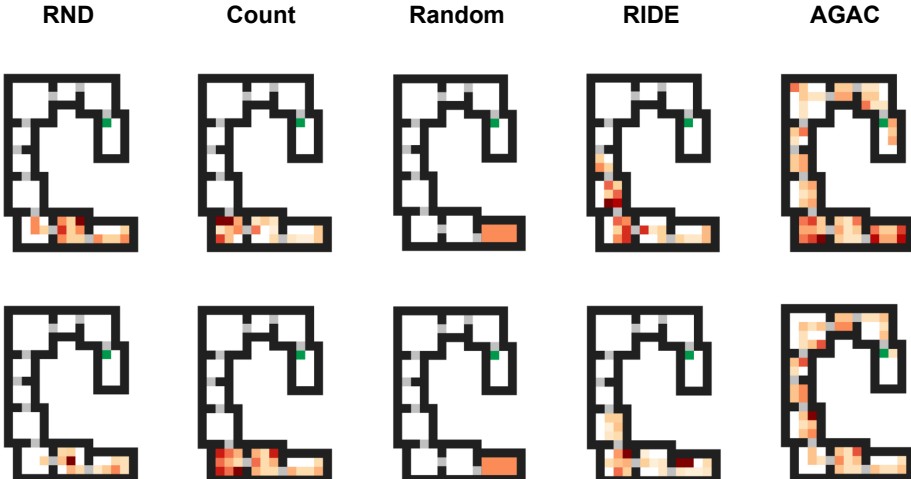

Figure 5: State visitation heatmaps for RND, Count, a random uniform policy, RIDE, and `AGAC` trained in a singleton environment (top row) and procedurally-generated environments (bottom row) without extrinsic reward for 10M timesteps in the MultiRoomN10S6 task.

## 5.4 Exploration in Reward-free Environment

To better understand the effectiveness of our method and inspect how the agent collects rewards that would not otherwise be achievable by simple exploration heuristics or other methods, we analyze the performance of `AGAC` in another (procedurally-generated) challenging environment, MultiRoomN10S6, when there is no reward signal, *i.e.* no extrinsic reward. Beyond the good performance of our method when extrinsic rewards are given to the agent, Fig. 4 indicates that the exploration induced by our method makes the agent succeed in a significant proportion of the episodes: in the configuration "NoExtrinsicReward" the reward signal is not given (the goal is invisible to the agent) and the performance of `AGAC` stabilizes around an average return of $\sim 0.15$. Since the return of an episode is either 0 or 1 (depending on whether the agent reached the goal state or not), and because this value is aggregated across several episodes, the results indicate that reward-free `AGAC` succeeds in $\sim 15\%$ of the tasks. Comparatively, random agents have a zero average return. This poor performance is in accordance with the results in Raileanu & Rocktäschel (2019) and reflects the complexity of the task: in order to go from one room to another, an agent must perform a specific action to open a door and cross it within the time limit of 200 timesteps. In the following, we visually investigate how different methods explore the environments.

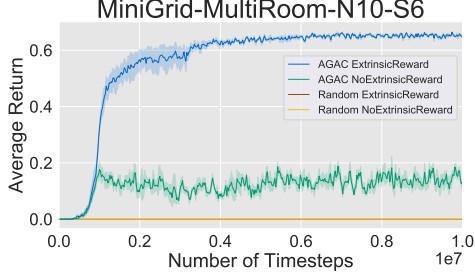

Figure 4: Average return on N10S6 with and without extrinsic reward.

## 5.5 Visualizing Coverage and Diversity

In this section, we first investigate how different methods explore environments without being guided by extrinsic rewards (the green goal is invisible to the agent) on both *procedurally-generated* and *singleton* environments. In singleton environments, an agent has to solve the same task in the same environment/maze in every episode. Fig. 5 shows the state visitation heatmaps (darker areas correspond to more visits) after a training of 10M timesteps. We observe that most of the methods explore inefficiently in a singleton environment and that only RIDE succeeds in reaching the fifth room while `AGAC` reaches the last (tenth) room. After training the agents in procedurally-generated environments, the methods explore even less efficiently while `AGAC` succeeds in exploring all rooms.

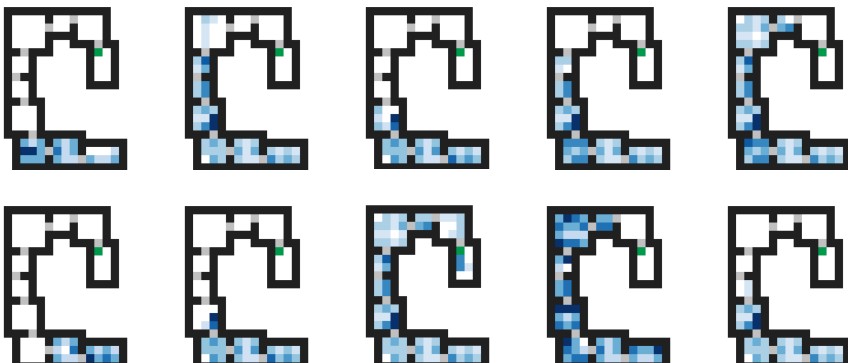

Figure 6: State visitation heatmaps of the last ten episodes of an agent trained in *procedurally-generated* environments without extrinsic reward for 10M timesteps in the MultiRoomN10S6 task. The agent is continuously engaging in new strategies.

We now qualitatively study the diversity of an agent's behavior when trained with `AGAC`. Fig. 6 presents the state visitation heatmaps of the last ten episodes for an agent trained in *procedurally-generated* environments in the MultiRoomN10S6 task without extrinsic reward. The heatmaps correspond to the behavior of the resulting policy, which is still learning from the `AGAC` objective. Looking at the figure, we can see that the strategies vary at each update with, for example, back-and-forth and back-to-start behaviors. Although there are no extrinsic reward, the strategies seem to diversify from one update to the next. Finally, Fig. 7 in Appendix A.2 shows the state visitation heatmaps in a different configuration: when the agent has been trained on a *singleton* environment in the MultiRoomN10S6 task without extrinsic reward. Same as previously, the agent is updated between each episode. Looking at the figure, we can make essentially the same observations as previously, with a noteworthy behavior in the fourth heatmap of the bottom row where it appears the agent went to the fourth room to remain inside it. Those episodes indicate that, although the agent sees the same environment repeatedly, the successive adversarial updates force it to continuously adapt its behavior and try new strategies.

## 6 DISCUSSION

This paper introduced `AGAC`, a modification to the traditional actor-critic framework: an adversary network is added as a third protagonist. The mechanics of `AGAC` have been discussed from a policy iteration point of view, and we provided theoretical insight into the inner workings of the proposed algorithm: the adversary forces the agent to remain close to the current policy while moving away from the previous ones. In a nutshell, the influence of the adversary makes the actor *conservatively diversified*.

In the experimental study, we have evaluated the adversarially-based bonus in VizDoom and empirically demonstrated its effectiveness and superiority compared to other relevant methods (some benefiting from count-based exploration). Then, we have conducted several performance experiments using `AGAC` and have shown a significant performance improvement over some of the most popular exploration methods (RIDE, AMIGo, Count, RND and ICM) on a set of various challenging tasks from MiniGrid. These procedurally-generated environments have served another purpose which is to validate the capacity of our method to generalize to unseen scenarios. In addition, the training stability of our method has been studied, showing a greater but acceptable sensitivity for $c$, the adversarial bonus coefficient. Finally, we have investigated the exploration capabilities of `AGAC` in a reward-free setting where the agent demonstrated exhaustive exploration through various strategic choices, confirming that the adversary successfully drives diversity in the behavior of the actor.

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

# A  ADDITIONAL EXPERIMENTS

## A.1  MINIGRID PERFORMANCE

In this section, we report the final performance of all methods considered in the MiniGrid experiments of Fig. 2 with the scores reported in Raileanu & Rocktäschel (2019) and Campero et al. (2021). All methods have a budget of 200M frames.

Table 2: Final average performance of all methods on several MiniGrid environments.

| Task | KC-S4R3 | KC-S5R3 | MR-N10S10 | OM-2Dlhb | OM-1Q | OM-2Q |
|------|---------|---------|-----------|----------|-------|-------|
| **AGAC** | **0.95** | **0.93** | **0.52** | **0.64** | **0.78** | **0.63** |
| RIDE | 0.19 | 0. | 0.40 | 0. | 0. | 0. |
| AMIGo | 0.54 | 0. | 0. | 0.20 | 0. | 0. |
| RND | 0. | 0. | 0. | 0.03 | 0. | 0. |
| Count | 0. | 0. | 0. | 0. | 0. | 0. |
| ICM | 0. | 0. | 0. | 0. | 0. | 0. |

## A.2  STATE VISITATION HEATMAPS IN SINGLETON ENVIRONMENT WITH NO EXTRINSIC REWARD

In this section, we provide additional state visitation heatmaps. The agent has been trained on a singleton environment from the MultiRoomN10S6 task without extrinsic reward. The last ten episodes of the training suggest that although the agent experiences the same maze over and over again, the updates force it to change behavior and try new strategies.

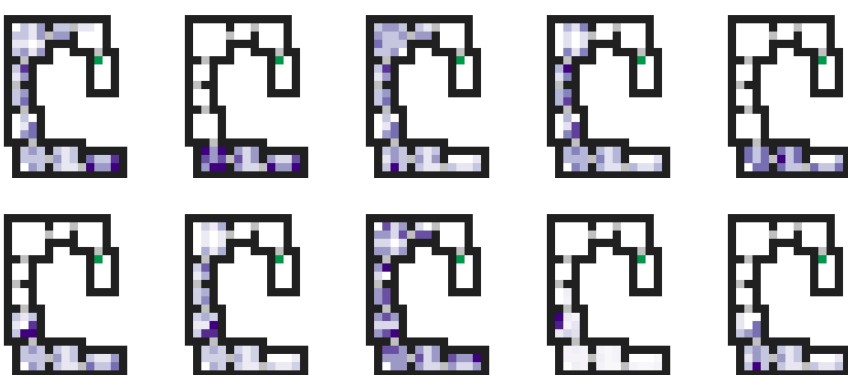

Figure 7: State visitation heatmaps of the last ten episodes of an agent trained in a *singleton* environment with no extrinsic reward 10M timesteps in the MultiRoomN10S6 task. The agent is continuously engaging into new strategies.

## A.3  (EXTREMELY) HARD-EXPLORATION TASKS WITH PARTIALLY-OBSERVABLE ENVIRONMENTS

In this section, we include additional experiments on one of the hardest tasks available in MiniGrid. The first is KeyCorridorS8R3, where the size of the rooms has been increased. In it, the agent has to pick up an object which is behind a locked door: the key is hidden in another room and the agent has to explore the environment to find it. The second, ObstructedMazeFull, is similar to ObstructedMaze4Q, where the agent has to pick up a box which is placed in one of the four corners of a 3x3 maze: the doors are locked, the keys are hidden in boxes and the doors are obstructed by balls. In those difficult tasks, only our method succeeds in exploring well enough to find rewards.

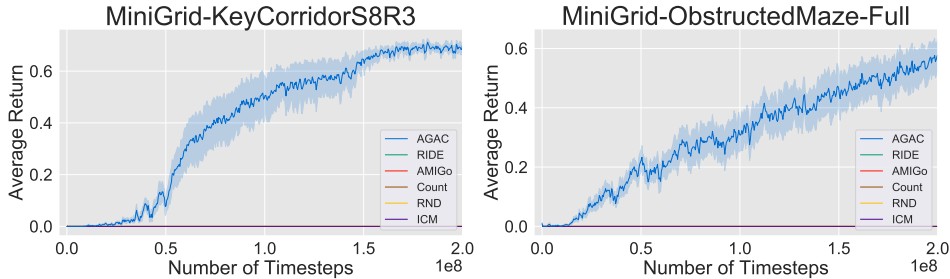

Figure 8: Performance evaluation of `AGAC` compared to RIDE, AMIGo, Count, RND and ICM on extremely hard-exploration problems.

### A.4 MEAN INTRINSIC REWARD

In this section, we report the mean intrinsic reward computed for an agent trained in Multi-RoomN12S10 to conveniently compare our results with that of Raileanu & Rocktäschel (2019). We observe in Fig. 9 that the intrinsic reward is consistently larger for our method and that, contrary to other methods, does not converge to low values. Please note that, in all considered experiments, the adversarial bonus coefficient $c$ in Eq. 2 and 3 is linearly annealed throughout the training since it is mainly useful at the beginning of learning when the rewards have not yet been met. In the long run, this coefficient may prevent the agent from solving the task by forcing it to always favour exploration over exploitation.

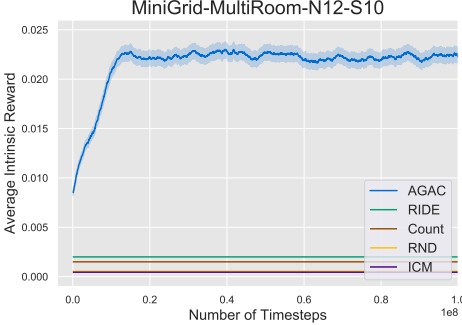

Figure 9: Average intrinsic reward for different methods trained in MultiRoomN12S10.

## B ILLUSTRATION OF `AGAC`

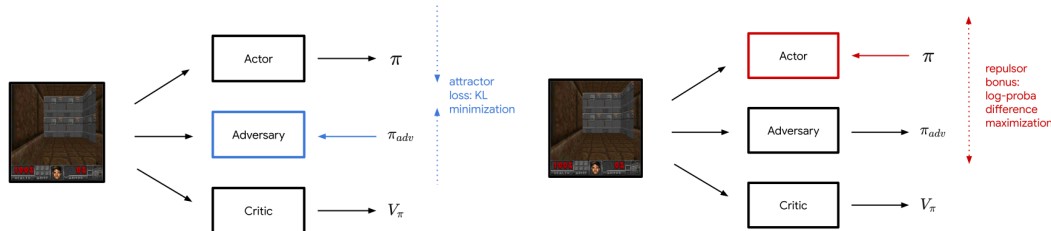

Figure 10: A simple schematic illustration of `AGAC`. **Left:** the adversary minimizes the KL-divergence with respect to the action probability distribution of the actor. **Right:** the actor receives a bonus when counteracting the predictions of the adversary.

## C EXPERIMENTAL DETAILS AND HYPERPARAMETERS

### C.1 MINIGRID SETUP

Here, we describe in more details the experimental setup we used in our MiniGrid experiments.

There are several different MiniGrid scenarios that we consider in this paper. MultiRoom corresponds to a set of navigation tasks, where the goal is to go from a starting state to a goal state. The notation MultiRoom-N2S4 means that there are 2 rooms in total, and that each room has a maximal side of 4. In order to go from one room to another, the agent must perform a specific action to open a door. Episodes are terminated with zero reward after a maximum of $20 \times N$ steps with $N$ the number of rooms. In KeyCorridor, the agent also has to pick up a key, since the goal state is behind a door that only lets it in with the key. The notation KeyCorridor-S3R4 means that there are 4 side corridors, leading to rooms that have a maximal side of 3. The maximum number of steps is 270. In ObstructedMaze, keys are hidden in boxes, and doors are obstructed by balls the agent has to get out of its way. The notation ObstructedMaze-1Dl means that there are two connected rooms of maximal side 6 and 1 door (versus a 3x3 matrix and 2 doors if the leading characters are $2D$), adding $h$ as a suffix places keys in boxes, and adding $b$ as a suffix adds balls in front of doors. Using $Q$ as a suffix is equivalent to using $lhb$ (that is, both hiding keys and placing balls to be moved). The maximum number of steps is 576. ObstructedMazeFull is the hardest configuration for this scenario, since it has the maximal number of keys, balls to move, and doors possible.

In each scenario, the agent has access to a partial view of the environment, a 7x7 square that includes itself and points in the direction of its previous movement.

### C.2 HYPERPARAMETERS

In all experiments, we train six different instances of our algorithm with different random seeds. In Table 3, we report the list of hyperparameters.

Table 3: Hyperparameters used in `AGAC`.

| Parameter | Value |
|---|---|
| Horizon $T$ | 2048 |
| Nb. epochs | 4 |
| Nb. minibatches | 8 |
| Nb. frames stacked | 4 |
| Nonlinearity | ELU (Clevert et al., 2016) |
| Discount $\gamma$ | 0.99 |
| GAE parameter $\lambda$ | 0.95 |
| PPO clipping parameter $\epsilon$ | 0.2 |
| $\beta_V$ | 0.5 |
| $c$ | $4 \cdot 10^{-4}$ ($4 \cdot 10^{-5}$ in VizDoom) |
| $c$ anneal schedule | linear |
| $\beta_{\mathrm{adv}}$ | $4 \cdot 10^{-5}$ |
| Adam stepsize $\eta_1$ | $3 \cdot 10^{-4}$ |
| Adam stepsize $\eta_2$ | $9 \cdot 10^{-5} = 0.3 \cdot \eta_1$ |

# D  IMPLEMENTATION DETAILS

In Fig. 11 is depicted the architecture of our method.

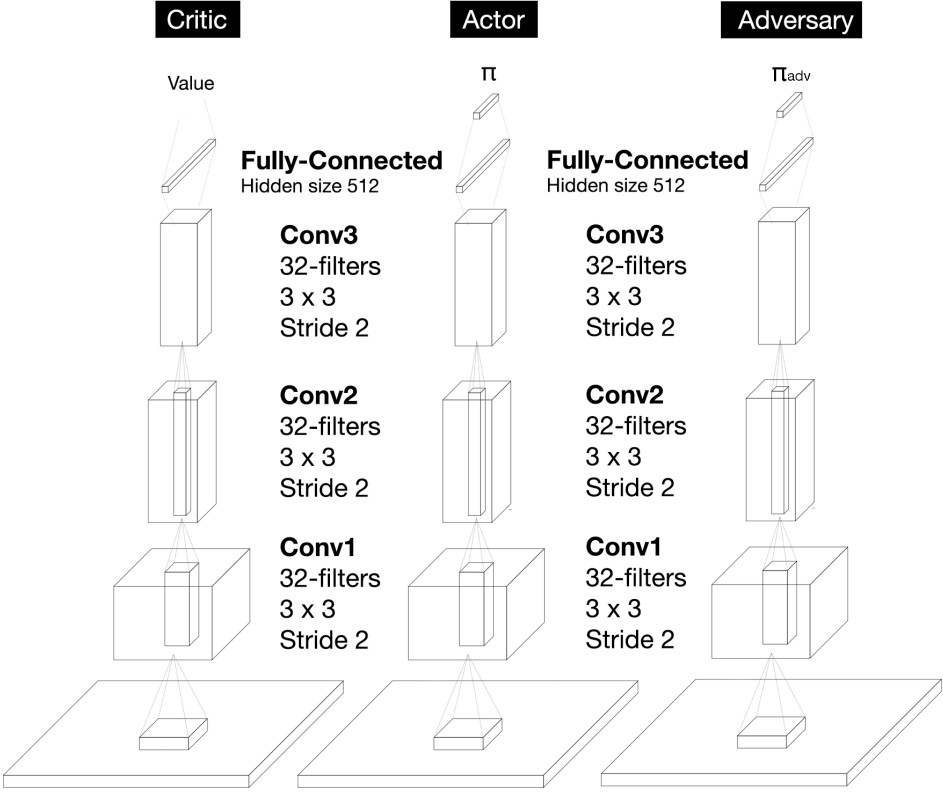

Figure 11: Artificial neural architecture of the critic, the actor and the adversary.

# E  PROOF OF SECTION 4.1 RESULTS

In this section, we provide a short proof for the result of the optimization problem in Section 4.1. We recall the result here:

$$\pi_{k+1} = \arg\max_\pi \mathcal{J}_{\mathrm{PI}}(\pi) \propto \left(\frac{\pi_k}{\pi_{\mathrm{adv}}}\right)^{\frac{c}{\alpha}} \exp\frac{Q_{\pi_k}}{\alpha},$$

with the objective function:

$$\mathcal{J}_{\mathrm{PI}}(\pi) = \mathbb{E}_s\mathbb{E}_{a\sim\pi(\cdot|s)}[Q_{\pi_k}(s,a) + c\left(\log\pi_k(a|s) - \log\pi_{\mathrm{adv}}(a|s)\right) - \alpha\log\pi(a|s)].$$

*Proof.* We first consider a simpler optimization problem: $\arg\max_\pi\langle\pi, Q_{\pi_k}\rangle + \alpha\mathcal{H}(\pi)$, whose solution is known (Vieillard et al., 2020a, Appendix A). The expression for the maximizer is the $\alpha$-scaled softmax:

$$\pi^* = \frac{\exp(\frac{Q_{\pi_k}}{\alpha})}{\langle 1,\, \exp(\frac{Q_{\pi_k}}{\alpha})\rangle}.$$

We now turn towards the optimization problem of interest, which we can rewrite as:

$$\arg\max_\pi\langle\pi, Q_{\pi_k} + c\left(\log\pi_k - \log\pi_{\mathrm{adv}}\right)\rangle + \alpha\mathcal{H}(\pi).$$

By the simple change of variable $\tilde{Q}_{\pi_k} = Q_{\pi_k} + c \left( \log \pi_k - \log \pi_{\mathrm{adv}} \right)$, we can reuse the previous solution (replacing $Q_{\pi_k}$ by $\tilde{Q}_{\pi_k}$). With the simplification:

$$\exp \frac{Q_{\pi_k} + c \left( \log \pi_k - \log \pi_{\mathrm{adv}} \right)}{\alpha} = \left( \frac{\pi_k}{\pi_{\mathrm{adv}}} \right)^{\frac{c}{\alpha}} \exp \frac{Q_{\pi_k}}{\alpha},$$

we obtain the result and conclude the proof. $\qquad\square$

