# OpenReview forum: "Adversarially Guided Actor-Critic"
_ICLR.cc/2021/Conference — ICLR 2021 Poster_

### Official Review · AnonReviewer1 · 2020-10-25
**This paper presented an interesting solution to actor-critic framework with adversarial learning**

**Rating:** 5
**Confidence:** 3

**Review:**

This paper proposed a new actor-critic framework with adversary guide for deep reinforcement learning (RL), and introduced new Kullback-Leiblier divergence bonus term based on the difference between actor network and adversary network to deal with the exploration in RL. The experimental results showed the merit of this method for exploration. Some comments are provided as follows.
1) Although the authors conducted analysis to carry out properties of hyperparameters, there are still something unclear in hyperparameter setting. The exploitation and exploration should be balanced during learning procedure. RL algorithms generally exploit more at the early stage and then explore at the later stage. But, this work fixed the exploration reward hyperparameter in learning procedure. Although this solution seems to be converged by fixing $c$, it should be better to implement a dynamic control scheme.
2) Section 5.5 showed the state vision heat maps to illustrate the capability of exploration using the proposed method. However, it would be convincing to make comparison over different methods.
3) Some equations were written in red font which should not be allowed in ICLR conference.

---

> ### Author Response · Authors · 2020-11-17
> **Response to Reviewer 1**
>
> We thank the reviewer for their comments.
>
> 1. We are on the same page with the reviewer regarding the exploration/exploitation trade-off. Actually, our method explicitly aims at balancing exploration and exploitation, as indicated at the end of Section 4.2: “In particular, the $c$ coefficient behind the adversarial bonus is linearly annealed”. As a result, the agent is most encouraged to escape the adversary’s predictions at the beginning of training, which leads to better exploration, and we reduce this incentive across time to avoid instability. While this is a simple dynamic control scheme, we find it to perform quite well in our experiments, and it does not introduce additional hyperparameters. Since this is a central point of the algorithm, we emphasize it more in the revised draft.
>
> 2. Thanks for suggesting to add heatmaps for different methods. We added heatmaps for several methods (uniform random policy, Count, RND, RIDE and AGAC) in Fig. 6 in the revised draft, for both fixed and procedurally-generated tasks. In a nutshell, we see that AGAC explores more efficiently and reaches further maze states than other methods, in both scenarios.
>
> 3. We do not quite understand the remark about the red font. We did that to emphasize the difference between our approach and a standard actor-critic. We think it makes the paper clearer. Is the problem using color or the red color specifically? We can easily change the color if necessary, maybe red is not the most convenient for color blind people. Could the reviewer expand on this?
>
> We hope to have addressed the reviewer's concerns. If this is the case, we invite the reviewer to consider updating their score. If not, additional comments would be welcomed.

---

### Official Review · AnonReviewer3 · 2020-10-27
**An adversarial extension of an actor critic model for efficient and generalisable exploration in very sparse reward environments with very competitive performance against sota methods.**

**Rating:** 7
**Confidence:** 2

**Review:**

The paper presents AGAC, an architecture for efficient, and generalisable, exploration in RL in settings with very sparse rewards. The model is compared against a number of SOTA methods for hard exploration problems on a number of procedurally generated environments, with very good performance results compared to the baselines.

The basic architecture extends an actor-critic model with an additional element, an adversary. The goal of the adversary is to predict correctly the actor's choices, minimizing its discrepancy from the actor. The goal of the actor, in addition to the standard maximization of expected return, is to maximize its discrepancy from the adversary, or in other words to stray away from its past self. The latter encourages exploration. AGAC quantifies the said discrepancy as the difference of the log propabilities of the actions under the actor and the adversary, the expectation of which under the actor is the KL divergence.

The actor-critic objective functions are adjusted as follows. The generalised advantage estimator contains now the discrepancy term, which encourages exploration. The critic's loss now includes as part of the target the KL divergence of the actor and the adversary. The adversary itself is trained to minimize the KL divergence from the actor.

The paper provides motivation for the design choises under a setting in which the policy loss is based on the Q value. Under such a setting the paper shows that the resulting objective, in addition to maximizing the return, it keeps the next update of the actor policy close to the previous actor and far from the adversary policy.

The experimental section includes a rich set of results over procedurally generated environments which evaluate how the exploration of AGAC does in unseen and very sparse reward environments. The evaluation results show rather important improvements over competive methods that seek to perform well in sparse reward environments.

I had some some clarity and presentation issues with the paper, see just bellow, overall this seems to be a simple idea which brings strong performance improvements in challenging settings.

Detailed questions:

With respect to the definition of the critic's objective function, eq 2. Does that
objective function derive naturally from the new definition of the generalised advantage,
eq 1? if yes a short explanation would be useful, if not what is the motivation for such
a target definition in learning the value function?

In presenting the motivation of AGAC in section 4.1, my understanding is that the adversary
is seen as a policy that represents the k-1, k-2, ... ? past policies. I would like to see
some more discussion on why is this so? Looking at the way the adversary is trained, eq 3,
I would say it tries to rather replicate the last, kth policy.

In the same section some more extensive discussion, maybe in the appendix, would be useful
when discussing the particular form of the solution for the policy iteration optimization
problem. In the discussion of the solution, what is \tau?

Section 5.2, hard exploration with partially observable polucy, I have a couple of terminology
issues.
* I am not sure what I should understand here by partially-observable policy? does that mean that the policy
has only access to a part of the environment/state description? as in a state-centric view? Wouldn't a better
term be partially-observable environments?
* In the same section the paper presents the intrinsic reward results, though in the case of AGAC
this has never been defined/described. I guess this refers to the exploration bonus, but it would
have been useful to clarify that.

Section 5.4 exploration with no reward.
* Probably a naive question: how do we see in fig 5 that the agent succeeds in a significant proportion of the episodes? is it the fact that we have an average non-zero return?
* I am not sure I see where the  confirmation of the fact that the agent exhaustively covers the environment comes from?


Section 5.5, diversity.
In figure 6 we see a still evolving policy, and I guess the bottom right heatmap of that figure is the final trained
agent. What is the difference from figure 7? I would have thought that in 7 we only see the trained agent, but then
the label speaks of "of the last ten episodes of an agent trained in a singleton environment" maybe the labels of the
figures got mixed?

---

> ### Author Response · Authors · 2020-11-17
> **Response to Reviewer 3**
>
> We thank the reviewer for their detailed comments.
>
> In AGAC, the objective function for the critic has indeed a natural origin. Adding the KL divergence to the critic target mirrors the action log-probability difference we add to the advantage, which can be interpreted nicely. Taking $\lambda \rightarrow 1$ for simplicity, under which the generalized advantage is $A_t = G_t - V_{\phi_{\text{old}}}(s_t)$, we can actually express the modified advantage as $A_{t}^{\text{AGAC}} = G_t - \hat{V}^{\phi_{\text{old}}}_{t} + c ( \log \pi(a_t | s_t) - \log \pi_\text{adv}(a_t | s_t)  - \hat{D}_{\mathrm{KL}}^{\phi_\text{old}}(\pi(\cdot | s_t)\|\|\pi_\text{adv}(\cdot | s_t)) )$ with $G_t$ the observed return, $\hat{V}^{\phi_\text{old}}_t$ the estimated return and $\hat{D}_\mathrm{KL}^{\phi_\text{old}}(\pi(\cdot | s_t)\|\|\pi_\text{adv}(\cdot | s_t))$ the estimated KL-divergence (both are estimated components of the modified critic learned in AGAC). What this decomposition shows is that AGAC favors transitions whose actions are less accurately predicted by the adversary than the average action, i.e. $\log \pi(a | s) - \log \pi_\text{adv}(a | s) \geq \hat{D}_\mathrm{KL}^{\phi_\text{old}}(\pi(\cdot | s)\|\pi_\text{adv}(\cdot | s))$. We included these details and revised the whole Section 4 in the updated draft, which we think is a lot clearer overall.
>
> In theory, $\pi_{adv}$ could indeed match the current policy $\pi_k$ since it gets supervision from $\pi_k$. We have several reasons to think it does not: it gets information with partial coverage about $\pi_k$, it has a limited optimization budget (few steps of SGD) and additionally we use a smaller learning rate to update $\pi_{adv}$. Thus, it is more realistic to consider that it matches an unknown mixture of all previous policies.
>
> We thank the reviewer for their suggestion of adding a small proof for the solution to the modified policy iteration scheme. The proof can be found in Appendix F in the revised draft. In a nutshell, we show that AGAC’s optimization problem belongs to a family of optimization problems (regularized policy iteration) whose closed-form solutions are known, and of which our solution is a simple variation.
>
> About Section 5.2, we indeed meant that the agent gets a partial view of the environment (a self-including 7x7 square, see Appendix B.1 for more information). We agree that partially-observable environments is a preferable terminology and employ it in the revised draft. We have also clarified the meaning of intrinsic reward in the context of AGAC, which is indeed the exploration bonus.
>
> Regarding Section 5.4, in Fig. 5 we see that the performance of reward-free AGAC stabilizes around an average return of ~0.15. Since the return of an episode is either 0 or 1 (depending on whether the agent reached the goal state or not), and since this value is aggregated across several episodes, it indicates that reward-free AGAC succeeds in ~15% of the procedurally-generated tasks. Comparatively, random agents have a zero average return, which we display in the updated Fig. 5. We rewrote unclear parts of Section 5.4 in the revised draft.
>
> About Section 5.5, they are right concerning Fig. 6: we present the last ten episodes of an agent trained in a procedurally-generated environment. For these ten episodes although, we fix the seed of the environment so that the layout remains the same, and the agent keeps learning. This allows us to study the evolution of behaviors across consecutive episodes. In Fig. 7, the agent is instead trained in a singleton environment (the layout is fixed, and the agent needs to solve the same task in the same environment at each episode). Hence the layout remains the same for each episode (not just the last ten). Note that in both experiments, there is no extrinsic reward because the goal of Section 5.5 is to investigate the capacity of our method to explore environments. We have reworked Section 5.5 to reflect their comments and also define “singleton” environments. We think the revised version provides a clearer description of the study.

---

### Official Review · AnonReviewer2 · 2020-10-27
**Good paper with minor issues in presentation (AnonReviewer2)**

**Rating:** 7
**Confidence:** 2

**Review:**

The authors propose a modification of the well-known actor-critic algorithm, give a intuition for how it works ("adding an adversary") and present experiments showing state-of-the-art performance on certain tasks on the VizDoom and MiniGrid environment, beating several recent baselines. While the improvements on previous algorithms are incremental, this is very much in line with recent papers in the field and certainly a worthwhile direction of research.

The paper appears well argued and is relatively well written, with only minor unclear parts and a few typos, listed below. Its results are impressive and I recommend publication.

As for areas of improvement, I didn't immediately understand all parts of the formulae in section 4 (and neither did I fully grasp the motivating simplification in section 4.1). For instance, it is not immediately obvious to me why the additional term in equation (4) could be described as a "bonus", as I see no reason the sign of $V_\phi(s_t)-\hat{V}_t$ couldn't be negative.

I would also propose reworking the figures. E.g., it's unclear to me where the comparison curves indicated in the label are in Figure 2. In various other figures, the baselines to compare to are indicated as step functions. I suspect this is because the raw data for these baselines wasn't available. Since the figures are trying to make a point about sample efficiency a table of numbers might not be the best alternative (although one might try that, perhaps giving the average return at several points during training?), but at least this reviewer isn't used to displaying individual data points as step functions of this sort.

The accompanying paragraph to Figure 2 also read somewhat mysteriously to this reviewer: "Results in Fig. 2 [...] indicate that AGAC clearly outperforms other methods in sample efficiency. Then, with nearly 2x more transitions, the graph in only ICM and RIDE match the score of AGAC." In combination with the lack of apparent ICM, RIDE, etc lines in Fig. 2 this makes for a confusing impression.

Certain other claims in the paper seem overblown or indeed irrelevant, to a greater extent yet than usual in the field of reinforcement learning. For example, take the statement "Note that in the configuration “NoExtrinsicReward”, the reward signal is not given. That is, the actor is not optimized for it, confirming that the agent exhaustively covers the environment." I'd propose to compare the performance here with that of a random agent, and perhaps also with a randomly sampled shallow neural network with sampling from output logits. The accompanying graph seems to be rather stable, as a random agent would be, and although it appears to show some improvement at the very beginning this may well be an artefact of the smoothing scheme used, or be otherwise postprocessing-related.

A similar comment goes for section 5.5, "Promoting Diversity". To this reviewer, this entire section lacks motivation for why a quasi-uniform heatmap would be exceptionally good? At least heatmaps generated by a random agent should be shown as a point of comparison -- might the agent simply be getting stuck randomly by a process analogous to a constrained brownian motion? In fact, what kind of behaviour of an agent trained solely on the intrinsic goal of not being predictable by one of its subsystems could be considered "good" vs. "bad" in the first place? The ultimate goal of "intrinsic motivation"-type agents like RIDE is to optimize an objective metric (a reward function, number of levels solved, specific states attained). Of course it's interesting to have an agent that's able to visit every state of the MDP, but it's not clear if that's hard in the given situation and doubtful that there was more than pure chance as a cause.

As an additional question, I was wondering if there are no shared parameters in the three parts of the model? As having shared parameters is common, it would be nice to mention this out explicitly.

Further typos:

"generalization in is a key challenge in RL"

Closing parenthesis in "Let $\pi : S \longrightarrow\Delta A)$"

Missing $\rm\LaTeX$ reference in "Fig. ??" on page 14 of the appendix.

Additionally, while I cannot claim to know the relevant literature exceptionally well, the claim "With the exception of Han & Sung (2020), which uses the entropy of the mixture between the policy induced from a replay buffer and the current policy as a regularizer, none of these methods explicitly use regularization to promote exploration." strikes me as dubious. For instance, Schmitt et al, "Kickstarting deep reinforcement learning" (2018), has a similar regularization scheme, which may "promote exploration" also in their case, as the mechanism of an algorithm is independent of its motivation or the particular angle used in its description.

All in all, this is a good paper which I enjoyed reading and I recommend it for publication in ICLR after some slight improvements.

---

> ### Author Response · Authors · 2020-11-17
> **Response to Reviewer 2**
>
> We thank the reviewer for their positive feedback and insightful comments. Below are replies to the questions the reviewer raised.
>
> Regarding the justification of the method (Section 4), we propose an updated version in the revised draft that we think makes the algorithm, high-level intuition and theoretical analysis clearer. For completion, we also include a short proof of the solution for the optimization problem of AGAC in the Policy Iteration setting. As for the specific terms in the equations being described as bonuses, we observe that in the RL literature the term “bonus” can be used for possibly negative quantities [1,2]. We consider the action log-probability difference as a bonus as it rewards diversified behavior, but it indeed could take negative values.
>
> We acknowledge that the step function figures (Fig. 2 and 3) are not ideal. While we cannot realistically run all methods and update all curves during the course of the discussion period, we do commit to including all regular curves in figures for the camera-ready version. In the meantime, we added Tables 1 and 2 that quantify more precisely the performance of the different methods and should make the accompanying paragraph to Fig. 2 less ambiguous.
>
> In the “NoExtrinsicReward” experiment, following their suggestion, we now report the performance of a uniform random policy (in Fig. 5). Note that, in this MiniGrid task, the agent must perform a specific action to open a door and go to the next room, and that episodes are limited to 200 timesteps of interaction, which in part explains why random agents never solve the task.
>
> We have also reworked Section 5.5 to reflect their comments and that of Reviewer 1. In particular, we add a new figure (Fig. 6) to the paper, which shows the state visitation heatmaps of a random agent in the same environment together with those of RND, Count, RIDE and AGAC. This new study provides an additional comparative perspective and further illustrates the difficulty of the task.
>
> Regarding the parameters of the model, there are no shared parameters between the three entities. This is indicated at the end of Section 4.2: “Note that the parameters are not shared between the policy, the critic and the adversary [...]”.
>
> We fixed all the typos that the reviewer identified.
>
> Finally, we edited the claim regarding the use of regularization for exploration (in Section 2), since we agree that it was not justified.
>
> [1] Savinov Nikolay, Anton Raichuk, Damien Vincent, Raphael Marinier, Marc Pollefeys, Timothy Lillicrap, and Sylvain Gelly. "Episodic Curiosity through Reachability." In International Conference on Learning Representations 2018.
> [2] Oudeyer Pierre-Yves, Frederic Kaplan, and Verena V. Hafner. "Intrinsic motivation systems for autonomous mental development." IEEE transactions on evolutionary computation 11, no. 2 (2007): 265-286.

---

### Author Response · Authors · 2020-11-24
**General response**

We would like to thank all the reviewers again for their insights and feedback. We have taken all comments into consideration, we did our best to address all concerns raised, and we think that the revised manuscript considerably improves the thoroughness of the experiments and clarity of writing.

Overall, we reworked Section 4 entirely to reflect the comments of R2 and R3: the high-level idea of AGAC, the elements of the algorithm, and its theoretical analysis were all reorganized for clarity. We incorporated a proof for the expression of the maximizer in the optimization problem of the Policy Iteration scheme for AGAC in Appendix F. We created additional tables that quantify the VizDoom and MiniGrid scores of all methods. As mentioned in our response to R2, we do commit to updating Fig. 2 and 3 with complete curves for all methods we compare against in the camera-ready version.

We have also added state visitation heatmaps in Fig. 6 to respond to the concerns raised by R1 and R2. These heatmaps visually assess the difference between the exploration strategy induced by our method and those of RIDE, RND, Count and a random uniform policy. In a nutshell, they show that our approach is the only one to reach the last (tenth) room in a complex reward-free task, which indicates that adversarially-induced diversity is good for exploration. Additionally, we have also enriched the set of tasks used to evaluate the performance of AGAC: the task MiniGrid-ObstructedMaze-2Q has been added to Fig. 3.

---

### Decision · Program_Chairs · 2021-01-07
**Final Decision**

**Decision:**

Accept (Poster)

**Comment:**

This work tackles to address the sparse reward problem in RL. They augment actor-critic algorithms by adding an adversarial policy. The adversary tries to mimic the actor while the actor itself tries to differentiate itself from the adversary in addition to learning to solve the task. This in a way provides diversity in exploration behavior. Reviewers liked the paper in general but had several clarification questions. The authors provided the rebuttal and addressed some of the concerns. Considering the reviews and rebuttal, AC and reviewers believe that the paper provides insights that are useful to share with the community. That being said, the paper will still immensely benefit with more extensive experimentation on standard benchmark environments like Atari, etc. Please refer to the reviews for other feedback and suggestions.